# A Novel E2 Glycoprotein Subunit Marker Vaccine Produced in Plant Is Able to Prevent Classical Swine Fever Virus Vertical Transmission after Double Vaccination

**DOI:** 10.3390/vaccines9050418

**Published:** 2021-04-22

**Authors:** Youngmin Park, Yeonsu Oh, Miaomiao Wang, Llilianne Ganges, José Alejandro Bohórquez, Soohong Park, Sungmin Gu, Jungae Park, Sangmin Lee, Jongkook Kim, Eun-Ju Sohn

**Affiliations:** 1BioApplications Inc., Pohang 37668, Korea; ympark86@postech.ac.kr (Y.P.); saysh13@bioapp.co.kr (S.P.); gusungmin@bioapp.co.kr (S.G.); oh@bioapp.co.kr (J.P.); lsm1978@bioapp.co.kr (S.L.); bio2020@bioapp.co.kr (J.K.); 2College of Veterinary Medicine and Institute of Veterinary Science, Kangwon National University, Chuncheon 24341, Korea; yeonoh@kangwon.ac.kr; 3OIE Reference Laboratory for Classical Swine Fever, IRTA-CReSA, 08193 Barcelona, Spain; miaomiao.wang@irta.cat (M.W.); llilianne.ganges@irta.cat (L.G.); josealejandro.bohorquez@irta.cat (J.A.B.)

**Keywords:** classical swine fever virus, E2 glycoprotein, marker vaccine, vertical transmission, plant, protection

## Abstract

The efficacy of a novel subunit vaccine candidate, based in the CSFV E2 glycoprotein produced in plants to prevent classical swine fever virus (CSFV) vertical transmission, was evaluated. A *Nicotiana benthamiana* tissue culture system was used to obtain a stable production of the E2-glycoprotein fused to the porcine Fc region of IgG. Ten pregnant sows were divided into three groups: Groups 1 and 2 (four sows each) were vaccinated with either 100 μg/dose or 300 μg/dose of the subunit vaccine at 64 days of pregnancy. Group 3 (two sows) was injected with PBS. Groups 1 and 2 were boosted with the same vaccine dose. At 10 days post second vaccination, the sows in Groups 2 and 3 were challenged with a highly virulent CSFV strain. The vaccinated sows remained clinically healthy and seroconverted rapidly, showing efficient neutralizing antibodies. The fetuses from vaccinated sows did not show gross lesions, and all analyzed tissue samples tested negative for CSFV replication. However, fetuses of non-vaccinated sows had high CSFV replication in tested tissue samples. The results suggested that in vaccinated sows, the plant produced E2 marker vaccine induced the protective immunogenicity at challenge, leading to protection from vertical transmission to fetuses.

## 1. Introduction

Classical swine fever (CSF) is considered as a highly contagious diseases affecting the Suidae family. The causative agent is a small, enveloped, single-stranded RNA virus known as CSF virus (CSFV), which belongs to the *Pestivirus* genus of the Flaviviridae family, together with several other viral species, including Bovine viral diarrhea virus and Border disease virus [1,2].

CSF outbreaks have major socio-economic consequences, including serious restrictions on international trade of pigs and pork-derived products [3]. The outbreak of this disease has also been accompanied by a high financial burden due to direct or indirect losses in the pig industry [4]. In addition, the ethical aspects linked to mass stamping out of herds in affected farms have made CSFV a notifiable pathogen by the World Organization for Animal Health (OIE). Although many countries manage this at the national level, the disease is endemic in several regions, such as South and Central America, the Caribbean and Asia [5,6].

Mass vaccination has been implemented in those countries as a mandatory control program for more than 50 years, mainly using live attenuated vaccines. In this regard, the C-strain has been identified as one of the most effective CSF vaccines, which broadly protects pigs from clinical CSF disease against all CSFV genotypes. However, the vaccination using these vaccines are immunologically indistinguishable from CSFV infection. Therefore, it is desirable to develop alternative vaccines such as subunit or modified virus-vectored vaccines, which allow differentiation of infected from vaccinated animals (DIVA) as well as having high efficacy [7].

The CSFV E2 glycoprotein is essential for viral replication and infection, and is the major immunogenic protein for inducing neutralizing antibodies to elicit protective immunity against CSFV [8,9]. Subunit vaccines based on the E2 of CSFV, developed years ago, have been regarded as a very suitable antigenic candidate for the DIVA vaccine.

Though scarcely used in field conditions, there are commercial DIVA vaccines based on the E2 protein, including recombinant E2 proteins expressed in insect cells [10,11,12], BAYOVAC CSF Marker (Bayer, Leverkusen, Germany) and Porcilis Pesti (Intervet, Boxmeer, Netherlands). Although some of those vaccines have been proven to prevent CSFV infection, there are reports to show limitations, especially in regard to early protection and transplacental transmission prevention [13,14]. Currently, a vaccine including the E2 fused with the CD154 (PorVac)(CIGB, Havana, Cuba), expressed in mammalian cells, is being used for disease control in the Caribbean region [15].

Safety and efficacy, as well as supply cost, are very important factors to consider when developing a vaccine for animals, especially livestock. In this respect, a plant-based manufacturing system is an attractive platform. In a previous study, a novel E2 glycoprotein of CSFV fused with the porcine Fc region of IgG was developed for increased stability in the host, and solubility [16]. Considering this, the main objective of the present study was to evaluate the effect of vaccination with this novel subunit marker vaccine produced in plant tissue, to prevent CSFV vertical transmission in pregnant sows.

## 2. Materials and Methods

### 2.1. Vaccine

We used a novel plant-produced E2 glycoprotein of CSFV subunit marker vaccine (E2-Bioapp) developed in a previous study [16]. Briefly, the pCAMBIA1300 MELCHE2 construct, containing cellulose binding domain-fused E2 recombinant protein, was prepared through digesting unnecessary domain, followed by ligation with the prepared pFc2 derived from porcine IgG—for better solubility and a longer half-life in the host— resulting in *pCAMBIA1300-pmE2:pFc2:HDEL*.

The fused construct, *pCAMBIA1300-pmE2:pFc2:HDEL,* was then introduced into *Agrobacterium tumefaciens* strain LBA4404, cultured in a *Nicotiana benthamiana* tissue culture system, harvested, and protein purification conducted. The purified and concentrated pmE2:pFc2 fusion protein was then adjuvanted in Emulsigen^®^-D adjuvant (MVP Adjuvant^®^, Omaha, NE, USA).

### 2.2. Cells and Viruses

Viral stocks were propagated and titrated in the porcine kidney cell line (PK-15, ATCC^®^:CCL-33, Manassas, VA, USA), cultured in Dulbecco’s Modified Eagle Medium (DMEM) supplemented with 10% fetal bovine serum pestivirus-free at 37 °C in 5% CO_2_. The highly virulent CSFV strain Margarita, which belongs to subgroups Genogroup 1.4 was employed for the challenge experiment and neutralization assay [17,18]. Additionally, the Alfort/187 strain of genotype 1.1, was also employed. Virus titration was conducted by end-point dilution using a peroxidase-linked assay (PLA) and the titers calculated according to Reed and Muench [19].

### 2.3. Experimental Design

At 57 days of gestation, 10 pregnant sows in 2~4 parity (Landrace × Yorkshire) were purchased, which tested negative for *Pestivirus*. All animals were analyzed again twice at six and two weeks before starting the study. All sows were housed throughout the experiment in an environmentally controlled building with pens over completely slatted floors. 

The sows were randomly divided into three groups. The sows in Group 1 (*n* = 4; Individual identification numbers are set from 1 to 4) were immunized with 100 µg of E2-Bioapp vaccine. The sows in Group 2 (*n* = 4; individual identification numbers are set from 5 to 8) were immunized with 300 µg of E2-Bioapp vaccine. The sows in Group 3 served as a non-vaccinated control group (*n* = 2; individual identification numbers are set from 9 to 10).

At 64 days of gestation, Groups 1 and 2 were inoculated intramuscularly on the neck with the E2-Bioapp vaccine, followed by boost immunization at 17 days post vaccination (dpv). At 10 days post second vaccination, the sows in Groups 2 and 3 were challenged intramuscularly on the neck with a 2.5 × 10^5^ tissue culture infective dose (TCID)/mL. On that date, sows in Group 1 were euthanized before viral challenge. Following the methodology used in previous studies, this experimental design did not include an extra group of animals vaccinated with commercial live attenuated CSFV vaccine [10,20,21,22]. Blood, nasal and rectal swab samples were collected at 0, 7, 17, 21 and 28 dpv, and at 4, 8, 14 and 18 days post challenge (dpc) to evaluate the CSFV-specific humoral immune response and virus shedding. At 18 dpc (109 days of gestation and around 2 weeks prepartum), the sows were euthanized, and gross examination was carried out in all fetuses [10]. Representative tissue samples including tonsil, spleen, thymus (only fetuses) and Peyer’s patch (only sows) in particular, were collected during necropsy. Following CSFV inoculation, the clinical signs in the sows were monitored daily by a trained veterinarian in a blinded manner, including rectal temperature. Animals were euthanized before the end of the trial if they presented clinical signs compatible with severe CSF or exhibited prostration behavior.

The experiment was evaluated and approved by the Ethical Committee of the Generalitat of Catalonia, Spain, according to existing European regulations (Project no. 1090).

### 2.4. Detection of CSFV E2-Specific and Neutralizing Antibodies

The serum samples were tested with a commercially available competitive CSFV Enzyme-Linked Immunosorbent Assay (ELISA) test (IDEXX Laboratories, Bern, Switzerland) for the detection of specific antibodies against the E2. In addition, a neutralization peroxidase-linked assay (NPLA) was also carried out. Neutralizing antibody titers were expressed as the reciprocal dilution of sera that neutralized 100TCID_50_ of Margarita or Alfort/187 strains in 50% of the culture replicates [23].

### 2.5. CSFV RNA Detection

Sera and nasal and rectal swabs, as well as tissue samples from sows and fetuses were subjected to RNA extraction for detection of CSFV RNA. The RNA extraction was performed using the IndiMag^®^ Pathogen Kit (Indical bioscience, Leipzig, Germany) according to the manufacturer’s instructions, and stored at −80 °C until use. Quantitative real-time (RT-qPCR) assay for the specific detection of CSFV RNA was performed in an ABI7500 instrument (Applied Biosystems, Foster City, CA, USA) [24]. Threshold cycle values (Ct) equal to, or above 40, were considered to be positive, whereas samples in which fluorescence was undetectable were considered to be negative. As previously described, Ct values above 28 were considered as low, from 23 to 28 as moderate and from 10 to 23, as high RNA viral load [25].

## 3. Results

### 3.1. E2-Antibody Response and Clinical Signs Generated in Sows after Vaccination and Challenge

At the time of vaccination (64 days of gestation), sows in all three groups were seronegative against CSFV. E2-specific antibodies started to appear at 7 dpv in one of the sows from Group 1 (Figure 1). All vaccinated sows had already seroconverted before the boost (17 dpv) indicating that single-dose vaccination can effectively induce E2-specific antibodies during 17 days.

At 21 dpv, the antibody response generated in the animals from Groups 1 and 2 was similar, while all the sows from Group 3 were negative. An increase in antibody response against E2 was observed in all the vaccinated sows after challenge and continued until the end of the experiment. By contrast, both sows in the non-vaccinated group showed fever, mild to moderate apathy, loss of appetite, weight loss, severe constipation, bloody stools and semi-prostration from day 4 after infection. These animals were euthanized at 10 dpc due to severe clinical signs associated with CSF. In addition, low antibody response against E2 was detected in only one of these animals, starting at 8 dpc (Figure 1).

### 3.2. Protection Levels against CSFV Replication in Vaccinated Sows after Challenge 

The vaccinated sows were protected from viremia and viral excretion throughout the trial. However, low viral RNA load was detected in some tissue samples from the vaccinated sows at 18 dpc. All of these animals exhibited viral RNA in tonsil tissue, whereas sows 5, 6, and 8 were also CSFV RNA positive in Peyer’s patch and spleen tissue. By contrast, CSFV RNA was detected in sera and nasal swabs from 4 dpc in non-vaccinated sows (Group 3). In addition, high viral RNA load was detected in all the clinical samples analyzed (sera, nasal and rectal swabs) (Figure 2), as well as in tissues (tonsil, Peyer’s patch and spleen) tested at 10 dpc (Figure 2).

### 3.3. Neutralizing Antibody Response Detected after Vaccination and Challenge

Neutralizing antibody response was present in all vaccinated sows from Group 2 before challenge (28 dpv). Meanwhile, absence of neutralizing activity was found in all sera samples collected from non-vaccinated sows in Group 3. The neutralizing activity against both Margarita (Table 1a) and Alfort/187 strains (Table 1b) detected in all vaccinated sows was progressively increased from 4 dpc until the end of the trial. The NPLA titers were slightly higher against Alfort/187 than those against Margarita strain. Conversely, non-vaccinated sows showed negative or very low NPLA titers against both strains after challenge (Table 1a,b).

### 3.4. Protection Levels of Fetuses from Either Vaccinated or Non-Vaccinated Sows after CSFV Challenge

Absence of gross lesions and mummifications were found in the litters from vaccinated sows (Group 2). All fetuses from these animals showed similar size and weight (Figure 3a). On the contrary, litters from unvaccinated sows (Group 3) showed different levels of mummifications (Figure 3b). Sow number 9 carried 18 fetuses, one of which was mummified and the rest were of irregular size, whereas sow 10 carried 20 fetuses, of which 7 were mummified and 13 were of irregular size. All fetuses from the non-vaccinated litters were CSFV RNA positive in one or more of the tissue samples tested (Table 2b). The fetuses of non-vaccinated sows showed obvious CSFV trans-placental transmissions with 84.6% and 92.3% of positivity. In direct contrast, the litters from vaccinated sows tested negative for CSFV replication by RT-qPCR in all samples analyzed, including serum, tonsil, spleen and thymus (Table 2a). 

## 4. Discussion

Despite significant efforts to control and eradicate CSF with mandatory vaccination policies, using live attenuated vaccines, the disease is still endemic. Sporadic outbreaks continue to occur in some countries [26,27]. Previous reports have suggested that the virus is conducted by an evolutionary process of circulating strains, due to the implementation of inefficient vaccination programs [27,28].

Low and moderate virulence CSFV strains have been reported to be circulating in pig herds. These strains have the capability to generate weak and persistently infected newborn piglets by trans-placental infection of the fetuses [29,30,31].

A novel E2 glycoprotein of CSFV subunit marker vaccine has been previously developed [16], which was produced in plant tissue and fused with the porcine Fc region of IgG. This production system fits the demand for vaccination in the veterinary field; being safe, effective and cost-affordable. The plant produced subunit marker vaccine candidate against CSFV was administered twice, and induced immunogenicity and protected pregnant sows from clinical signs, as well as reproductive failure. However, animals of Group 1 were not challenged, since before the challenge the antibody titers were similar to those of Group 2.

Neutralizing antibodies (NA) provide the best evidence that protective immunity has been established. The close correlation between NA titer to protection has been widely reported. It is accepted that a reasonable threshold antibody level for protection is 1:32 dilution of serum by NPLA test [32]. The NPLA titers increased rapidly and were high in vaccinated sows, and were shown to protect the animals from the development of CSF disease. Moreover, the vaccination aided in controlling viral replication after challenge, protecting the animals from developing viremia and viral shedding. Interestingly, the NPLA titers before challenge were only present against Alfort/187 strain among vaccinated groups (Group 1 and 2). It might be explained by the fact that the E2 protein of the vaccine formulation has a similar portion with the native fraction of Alfort/187 strain. Likewise, the antibody response generated by the E2-Bioapp vaccine was shown to have neutralizing activity against genotype 1.4 (Margarita) as well as genotype 1.1 (Alfort/187), suggesting the capacity of E2-Bioapp to confer protection against heterologous strains.

However, viral RNA was still detected in lymphoid tissue of vaccinated sows when all other tissues were cleared. The virus was inoculated intramuscularly to the sows that were previously vaccinated, being retained in lymphoid tissue. Since CSFV has a tropism for the lymphoid system, it may explain why the virus is detected in these tissues [33]. Possibly, it would take more than the 18 days to clear viral RNA in lymphoid tissue due to the viral tropism to this tissue.

Notably, fetuses from vaccinated sows were protected from CSFV transplacental transmission, shown by the absence of CSFV RNA measured by RT-qPCR. It is well established that trans-placental transmission of CSFV is generated mainly during mid-gestation [30,34,35]. The outcome of trans-placental infection of fetuses depends largely on the time of gestation and viral virulence [35]. CSFV trans-placental transmission may result in abortion, stillbirth, mummification, malformations or the birth of weak persistently infected piglets [36].

The glycosylation of E2 protein produced in plant tissue was confirmed elsewhere [16] and correct glycosylation of antigenic proteins has been known to affect antigen efficacy [37]. However, the effect of xylose and fucose epitopes on N-glycans produced in plants is controversial and how plant-specific glycans on E2 protein affect antigenicity of the subunit vaccine in pigs remains elusive. 

Collectively, the results of the present study support the capacity of E2-Bioapp vaccine to protect against transplacental transmission from the sow to the fetuses after two vaccination doses. Further studies will provide insight to elucidate the efficacy of this vaccine prototype in different age groups. The E2-Bioapp vaccine candidate is an economically feasible strategy, applicable for large scale production in order to guarantee vaccine coverage in large populations. This work opens the window for the development of new strategies for vaccine production in animal health.

## 5. Conclusions

E2 glycoprotein subunit marker vaccine produced in plant tissue generated a rapid and high antibody response against the E2 glycoprotein. In addition, it is able to safely induce neutralizing antibody in sows, and subsequently protect against vertical transmission of swine fever virus challenge.

## Figures and Tables

**Figure 1 vaccines-09-00418-f001:**
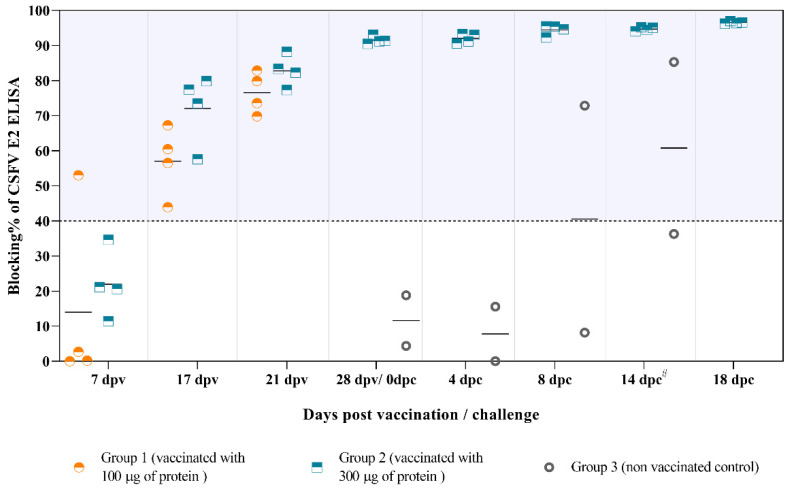
The antibody responses against the E2 glycoprotein after vaccination and challenge. The E2-specific antibody response was analyzed by ELISA and represented as blocking percentage (blocking %). Blocking % values equal to or greater than 40% were considered as positive. **^#^**: Sows 9 and 10 were euthanized at 10 dpc due to severe clinical signs and the data are represented at this time-point.

**Figure 2 vaccines-09-00418-f002:**
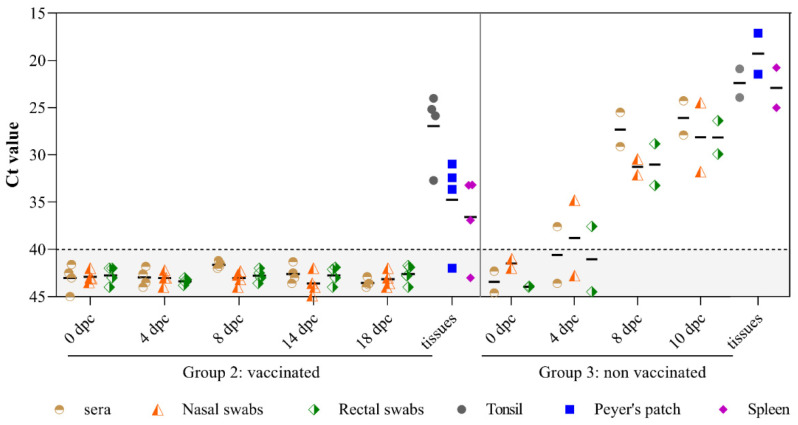
CSFV RNA detection in tissue samples from either vaccinated or non-vaccinated sows after challenge. CSFV RNA in sera, nasal and rectal swabs and tissue samples including tonsil, Peyer’s patch and spleen from the challenged sows was assessed by qRT-PCR. Ct values less than 40 were considered positive.

**Figure 3 vaccines-09-00418-f003:**
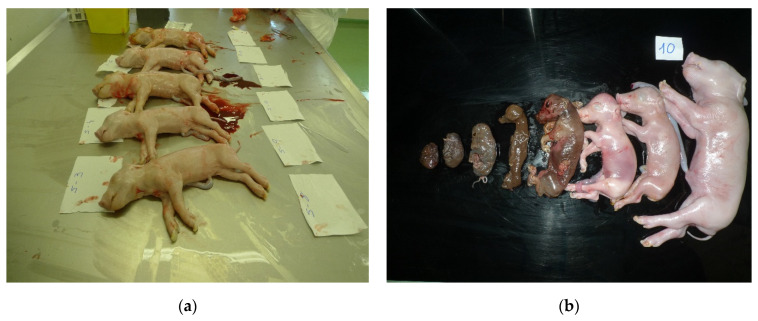
Representative fetuses of sows challenged with highly virulent CSFV Margarita strain. (**a**) Litters of sows vaccinated with E2-Bioapp (300 µg/dose) were observed healthy and uniform in size. (**b**) Fetuses from the unvaccinated sows were mummified and the remaining showed uneven size.

**Table 1 vaccines-09-00418-t001:** CSFV neutralizing peroxidase linked antibody (NPLA) titer: (**a**) NPLA titer to CSFV strain Margarita, (**b**) NPLA titer to CSFV strain Alfort/187.

**(a)**									
**NPLA to Margarita Strain (Challenge Virus)**									
**Sow ID**	**Days Post Vaccination (dpv)**				**Days Post Challenge (dpc)**				
	**0 dpv**	**7 dpv**	**17 dpv**	**21 dpv**	**28 dpv/0 dpc**	**4 dpc**	**8 dpc**	**14 dpc ***	**18 dpc**
Group 1: Sows inoculated with 100 µg of protein									
Sow 1	Neg **	Neg	Neg	1/10					
Sow 2	Neg	Neg	Neg	1/20					
Sow 3	Neg	Neg	Neg	Neg					
Sow 4	Neg	Neg	Neg	Neg					
Group 2: Sows inoculated with 300 µg of protein									
Sow 5	Neg	Neg	Neg	1/10	1/40	1/80	1/1280	1/2560	1/1280
Sow 6	Neg	Neg	Neg	Neg	1/40	1/20	1/320	1/2560	1/1280
Sow 7	Neg	Neg	1/10	1/40	1/160	1/320	1/320	1/2560	1/2560
Sow 8	Neg	Neg	Neg	Neg	1/40	1/20	1/160	1/2560	1/640
Group 3: Non vaccinated control animals									
Sow 9	Neg	Neg	Neg	Neg	Neg	Neg	Neg	1/10	
Sow 10	Neg	Neg	Neg	Neg	Neg	Neg	Neg	Neg	
**(b)**									
**NPLA to Alfort 187 CSFV Strain**									
**Sow ID**	**Days Post Vaccination (dpv)**				**Days Post Challenge (dpc)**				
	**0 dpv**	**7 dpv**	**17 dpv**	**21 dpv**	**28 dpv/0 dpc**	**4 dpc**	**8 dpc**	**14 dpc ***	**18 dpc**
Group 1: Sows inoculated with 100 µg of protein									
Sow 1	Neg **	Neg	1/10	1/40					
Sow 2	Neg	Neg	1/10	1/320					
Sow 3	Neg	Neg	1/10	1/20					
Sow 4	Neg	Neg	1/20	1/20					
Group 2: Sows inoculated with 300 µg of protein									
Sow 5	Neg	Neg	Neg	1/10	1/1280	1/320	1/5120	1/10240	1/10240
Sow 6	Neg	Neg	Neg	1/10	1/160	1/160	1/5120	1/5120	1/5120
Sow 7	Neg	Neg	1/20	1/160	1/2560	1/2560	1/5120	1/10240	1/10240
Sow 8	Neg	Neg	1/10	1/20	1/640	1/160	1/640	1/5120	1/2560
Group 3: Non vaccinated control animals									
Sow 9	Neg	Neg	Neg	Neg	Neg	Neg	1/10	1/10	
Sow 10	Neg	Neg	Neg	Neg	Neg	Neg	Neg	Neg	

* Sow 9 and 10 were measured at 10 dpc, and for ethical reasons, were euthanized due to severe clinical symptoms. ** Neg means negative.

**Table 2 vaccines-09-00418-t002:** Detection of CSFV RNA in tissue samples of fetuses from either vaccinated or non-vaccinated sows after challenge: (**a**) Tissue samples of fetuses from vaccinated sows, (**b**) Tissue samples of fetuses from non-vaccinated sows.

**(a)**									
**Fetus ID**	**CSFV RT-qPCR (Ct Value)**				**Fetus ID**	**CSFV RT-qPCR (Ct Value)**			
	**Serum**	**Tonsil**	**Spleen**	**Thymus**		**Serum**	**Tonsil**	**Spleen**	**Thymus**
Fetuses from sow 5					Fetuses from sow 7				
1	Neg *	Neg	Neg	Neg	1	Neg	Neg	Neg	Neg
2	Neg	Neg	Neg	Neg	2	Neg	Neg	Neg	Neg
3	Neg	Neg	Neg	Neg	3	Neg	Neg	Neg	Neg
4	Neg	Neg	Neg	Neg	4	Neg	Neg	Neg	Neg
5	Neg	Neg	Neg	Neg	5	Neg	Neg	Neg	Neg
6	Neg	Neg	Neg	Neg	6	Neg	Neg	Neg	Neg
7	Neg	Neg	Neg	Neg	7	Neg	Neg	Neg	Neg
8	Neg	Neg	Neg	Neg	8	Neg	Neg	Neg	Neg
9	Neg	Neg	Neg	Neg	9	Neg	Neg	Neg	Neg
10	Neg	Neg	Neg	Neg	10	Neg	Neg	Neg	Neg
11	Neg	Neg	Neg	Neg	11	Neg	Neg	Neg	Neg
12	Neg	Neg	Neg	Neg	12	Neg	Neg	Neg	Neg
13	Neg	Neg	Neg	Neg	13	Neg	Neg	Neg	Neg
Fetuses from sow 6					Fetuses from sow 8				
1	Neg	Neg	Neg	Neg	1	Neg	Neg	Neg	Neg
2	Neg	Neg	Neg	Neg	2	Neg	Neg	Neg	Neg
3	Neg	Neg	Neg	Neg	3	Neg	Neg	Neg	Neg
4	Neg	Neg	Neg	Neg	4	Neg	Neg	Neg	Neg
5	Neg	Neg	Neg	Neg	5	Neg	Neg	Neg	Neg
6	Neg	Neg	Neg	Neg	6	Neg	Neg	Neg	Neg
7	Neg	Neg	Neg	Neg	7	Neg	Neg	Neg	Neg
-	-	-	-	-	8	Neg	Neg	Neg	Neg
-	-	-	-	-	9	Neg	Neg	Neg	Neg
-	-	-	-	-	10	Neg	Neg	Neg	Neg
-	-	-	-	-	11	Neg	Neg	Neg	Neg
-	-	-	-	-	12	Neg	Neg	Neg	Neg
-	-	-	-	-	13	Neg	Neg	Neg	Neg
**(b)**									
**Fetus ID**	**CSFV RT-qPCR (Ct Value)**				**Fetus ID**	**CSFV RT-qPCR (Ct Value)**			
	**Serum**	**Tonsil**	**Spleen**	**Thymus**		**Serum**	**Tonsil**	**Spleen**	**Thymus**
Fetuses from sow 9					Fetuses from sow 10				
1	Neg *	Neg	36.50	Neg	1	Neg	35.58	Neg	36.56
2	Neg	Neg	36.60	Neg	2	Neg	31.22	26.63	23.78
3	33.39	30.78	24.44	23.52	3	Neg	Neg	30.48	Neg
4	Neg	Neg	Neg	34.51	4	Neg	33.90	30.36	27.99
5	Neg	39.83	Neg	29.71	5	Neg	37.23	31.91	32.03
6	Neg	31.89	35.00	Neg	6	Neg	32.46	Neg	Neg
7	Neg	Neg	35.15	Neg	7	Neg	34.40	Neg	36.50
8	Neg	36.00	Neg	Neg	8	Neg	34.20	Neg	35.65
9	Neg	Neg	Neg	Neg	9	Neg	Neg	Neg	30.28
10	Neg	37.85	Neg	35.80	10	Neg	33.85	27.17	29.16
11	Neg	34.46	30.96	26.82	11	Neg	Neg	Neg	30.43
12	Neg	Neg	Neg	Neg	12	Neg	29.85	36.44	33.47
13	Neg	30.64	28.92	29.56	13	Neg	32.99	27.05	26.39

* Neg means negative.

## Data Availability

The data presented in this study are available on request from the corresponding author.

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
