# Peer review of "A Novel E2 Glycoprotein Subunit Marker Vaccine Produced in Plant Is Able to Prevent Classical Swine Fever Virus Vertical Transmission after Double Vaccination"

_vaccines, 2021, doi:10.3390/vaccines9050418_

Round 1

Reviewer 1 Report

This original article describes the systematic evaluation of a novel subunit vaccine against classical swine fever virus and its ability to protect against vertical transmission of virus to fetuses. The vaccine - a fusion protein of the CSTV-E2-glycoprotein and porcine IgG-Fc is very well designed for optimum pharmacokinetics and prolongened plasma half life. In addition, the subunit vaccine has been elegantly manufactured using Nicotiana tabacum as a production host.

The study is very well designed and yielded very informative data. The authors clearly provided evidence that this subunit vaccine is successful in providing complete immunity to vaccinated sows and their offspring. Neutralizing peroxidase linked assay data showed that Neutralizing Antibody titers increased fast and high in vacinated sows. This is all the more exciting since a subunit vaccine like this is likely to have desirable DIVA properties - allowing veterinarians to differentiate vaccinated from infected animals in areas where swine fever is endemic. Last but not least the study also contains very informative data about suitable dosing strength and vaccination intervals.

Although the article is really well written in a short and no-nonsense style, it provides the reader with only a rather short and abbreviated discussion of the data. This leaves a couple of questions open to the interested reader:

Methods section:

Why were the virus titers for the virulent challenging strains calculated by the age-old method of Reed & Muench - first published in 1938? Wouldn’t it have been more accurate to calculate the inflection point for the TCID50-end-point-dilution assay with OriginPro or Graph Pad Prism or some other analysis and graphing solution software or is the R&M method just grandfathered in?

Discussion section:

Has the structure of the plant-derived glycans attached to the subunit vaccine been elucidated? Did this struture in any way contribute to vaccine efficacy or is it even likely that it did? It would be nice to read more about that in the discussion.

How could it be explained that one animal in the non-vaccinated group in Figure 1 developed antibodies post challenge while the others didn’t? What kind of antibodies where these? IgM, IgG?

Figure 2: How could it be explained that viral RNA above ct-threshold level was still detected in lymphatic tissue of vaccinated sows but not in other tissues? Would be nice to read about this in the discussion.

Table1: What is the reason why there are no data presented for group 1 sows post challenge?

In any case the article provides very nice data on a new subunit vaccine of commercial importance and I therefore recommend to accept it for publication.

Author Response

Dear reviewer,

We did our best to respond to your comments.

Point 1

Methods section:

Why were the virus titers for the virulent challenging strains calculated by the age-old method of Reed & Muench - first published in 1938? Wouldn’t it have been more accurate to calculate the inflection point for the TCID50-end-point-dilution assay with OriginPro or Graph Pad Prism or some other analysis and graphing solution software or is the R&M method just grandfathered in?

Response 1: The method is widely used in the field of virology for the calculation of the viral titer. Likewise, in one of the methods accepted by the manual of the world organization for animal health (OIE) that regulates the isolation and titration techniques for CSFV that must be carried out worldwide. In addition, the method used by our working group to titrate CSFV (in addition to other virology research groups in the world), see a short list of references from high impact factor journals with different viruses of various viral families, in which this method has been used (not by old means that it is out of date):

List of references:

  • Decrypting the Origin and Pathogenesis in Pregnant Ewes of a New Ovine Pestivirus Closely Related to Classical Swine Fever Virus.
  • Wang M, Sozzi E, Bohórquez JA, Alberch M, Pujols J, Cantero G, Gaffuri A, Lelli D, Rosell R, Bensaid A, Domingo M, Pérez LJ, Moreno A, Ganges L.Viruses. 2020 Jul 17;12(7):775. doi: 10.3390/v12070775.
  • Differential susceptibility & replication potential of Vero E6, BHK-21, RD, A-549, C6/36 cells & Aedes aegypti mosquitoes to three strains of chikungunya virus. Sudeep AB, Vyas PB, Parashar D, Shil P. Indian J Med Res. 2019 Jun;149(6):771-777. doi: 10.4103/ijmr.IJMR_453_17. A rapid method for titration of ascovirus infectivity. Han N, Chen Z, Wan H, Huang G, Li J, Jin BR. J Virol Methods. 2018 May;255:101-106. doi: 10.1016/j.jviromet.2018.02.011. Epub 2018 Feb 14.
  • Lithium chloride inhibits early stages of foot-and-mouth disease virus (FMDV) replication in vitro. Zhao FR, Xie YL, Liu ZZ, Shao JJ, Li SF, Zhang YG, Chang HY. J Med Virol. 2017 Nov;89(11):2041-2046. doi: 10.1002/jmv.24821. Epub 2017 Aug 28.
  • T4 bacteriophage-mediated inhibition of adsorption and replication of human adenovirus in vitro. Przybylski M, Borysowski J, Jakubowska-Zahorska R, Weber-Dąbrowska B, Górski A. Future Microbiol. 2015;10(4):453-60. doi: 10.2217/fmb.14.147.

Discussion section:

Point 2: Has the structure of the plant-derived glycans attached to the subunit vaccine been elucidated? Did this structure in any way contribute to vaccine efficacy or is it even likely that it did? It would be nice to read more about that in the discussion.

Response 2: We mentioned glycosylation of E2 produced in plants in Discussion section (lines 249-253) as below,

“The glycosylation of E2 protein produced in plant was confirmed in elsewhere [16] and correct glycosylation of antigenic proteins has been known to affect antigen efficacy [37]. However, the effect of xylose and fucose epitopes on N-glycans produced in plants is controversial and how plant-specific glycans on E2 protein affect antigenicity of the subunit vaccine in pigs remains to be elusive.”

Point 3: How could it be explained that one animal in the non-vaccinated group in Figure 1 developed antibodies post challenge while the others didn’t? What kind of antibodies where these? IgM, IgG?

Response 3: The difference in the levels of specific antibodies against E2 glycoprotein (this ELISA quantifies porcine IgG, specified in the kit instructions) detected in sows 9 and 10 (both from the control group), is explained by the individual genetic variability of the animals used in the study. This is a frequent behavior in adult animals after infection with CSFV, some are positive for total antibodies against the E2 protein before other animals. However, the results of this work show a quite important difference, correctly reflected in Figures 1 and Tables 1a and 1b, between the sows of the control group (3) and the vaccinated sows, both after vaccination and viral challenge. Thus, a positive effect on the response of vaccinated sows in terms of their capacity for protection mediated by antibodies is demonstrated.

Likewise, it should be noted that, as explained in the discussion, it is the neutralizing antibody titers that correlate with protection against classical swine fever virus. In this sense, both sows in control group 3 have a similar behavior since between 8 and 10 days post infection they are both NEGATIVE, one of them being a very weak positive with a low titer of 1:10 (which is not relevant in correlating with protective capacity) and is practically irrelevant.

Point 4: Figure 2: How could it be explained that viral RNA above ct-threshold level was still detected in lymphatic tissue of vaccinated sows but not in other tissues? Would be nice to read about this in the discussion.

Response 4: We mentioned it In Discussion section (Lines 236-241) as below,

The virus was inoculated intramuscularly to the sows that were previously vaccinated, being retained in lymphoid tissue (since the animals were protected from viremia). CSFV has a tropism for the lymphoid system, which explains why the virus is detected in these tissues. Possibly, it would take more time than the 18 days that these animals were left in the study after the viral challenge to observe that little by little the RNA is disappearing. Further studies will be required to optimize the dose in pregnant sows to reduce this time.

Point 5: Table 1: What is the reason why there are no data presented for group 1 sows post challenge?

Response 5: These animals were not challenged (group 1), since before the challenge the antibody titers were similar to those of group 2. It was mentioned in Discussion section (Lines 221-222)

Reviewer 2 Report

The authors deal with very interesting topic: development of an effective E2 glycoprotein subunit marker vaccine to prevent swine fever virus.

The manuscript is, in principle, suitable for publication, but it needs some improvements prior to that.

  1. Introduction, paragraph 1: Family names do not need to be italic. e.g. Suidae, the Flaviviridae
  2. Lines 43- 47: please add references.
  3. Line 141: Please spell out
  4. Figure 1: Data points are not aligned with 0, 7, 17, 21 and 28 dpv, and 4, 8, 14 and 18 dpc.
  5. Was there any specific reason for not challenging group 1 sows?
  6. Even through this work did not include an extra group of commercial live attenuated CSFV vaccine, could you still compare your results with any previously reported data of commercial live attenuated CSFV vaccine?

Author Response

Dear Reviewer,

We did our best to respond to your comments.

Point 1: Introduction, paragraph 1: Family names do not need to be italic. e.g. Suidae, the Flaviviridae

Response 1: We revised as you commented.

Point 2: Lines 43- 47: please add references.
Response 2: We added references.

Point 3: Line 141: Please spell out

Response 3: We have provided additional explanations for this sentence as follows:

“All vaccinated sows had already seroconverted before boost (17 dpv) indicating that single-dose vaccination can effectively induce E2-specific antibodies during 17 days.”

Point 4: Figure 1: Data points are not aligned with 0, 7, 17, 21 and 28 dpv, and 4, 8, 14 and 18 dpc.

Response 4: While what the reviewer says is true, the goal of constructing this figure is not to present the data aligned. We consider that the results are clearly represented in Figure 1 clearly reflecting the kinetics and differences between groups. The same groups are not presented at all times. Therefore, the style of the figure could not be the same if we align it. Thank you for your comment.

Point 5: Was there any specific reason for not challenging group 1 sows?

Response 5: Answer: These animals were not challenged (group 1), since before the challenge the antibody titers were similar to those of group 2. It was mentioned in Discussion section (Lines 221-222).

Point 6: Even through this work did not include an extra group of commercial live attenuated CSFV vaccine, could you still compare your results with any previously reported data of commercial live attenuated CSFV vaccine?

Response 6: Unfortunately, there is no article with a similar experimental design that can be comparable.

It is not possible to make any comparison without reference to previous published data with the efficacy of the Chinese strain, practicing vaccination in the third of the gestation in which it has been carried out here and also, making a viral challenge such as the one practiced in this study.

Although vaccination with the Chinese strain is widespread, there are few scientific articles on the efficacy of this vaccine to prevent transplacental transmission of CSFV, that is why it is not referred to in the discussion and it was not the objective of this study to make a comparison side by side with the Chinese strain,

Few articles on the C strain in pregnant sows:

  • Classical swine fever virus strain "C" protects the offspring by oral immunisation of pregnant sows.Kaden V, Lange E, Steyer H, Lange B, Klopfleisch R, Teifke JP, Bruer W.Vet Microbiol. 2008 Jul 27;130(1-2):20-7. doi: 10.1016/j.vetmic.2007.12.022.

  • Protection of gruntlings against classical swine fever virus-infection after oral vaccination of sows with C-strain vaccine. Kaden V, Lange E, Müller T, Teuffert J, Teifke JP, Riebe R.J Vet Med B Infect Dis Vet Public Health. 2006 Dec;53(10):455-60. doi: 10.1111/j.1439-0450.2006.00993.x.PMID: 17123422